

# GASTRO-CADx: a three stages framework for diagnosing gastrointestinal diseases

Omneya Attallah and Maha Sharkas

Department of Electronics and Communication Engineering, College of Engineering and Technology, Arab Academy for Science, Technology and Maritime Transport, Alexandria, Egypt

## ABSTRACT

Gastrointestinal (GI) diseases are common illnesses that affect the GI tract. Diagnosing these GI diseases is quite expensive, complicated, and challenging. A computer-aided diagnosis (CADx) system based on deep learning (DL) techniques could considerably lower the examination cost processes and increase the speed and quality of diagnosis. Therefore, this article proposes a CADx system called Gastro-CADx to classify several GI diseases using DL techniques. Gastro-CADx involves three progressive stages. Initially, four different CNNs are used as feature extractors to extract spatial features. Most of the related work based on DL approaches extracted spatial features only. However, in the following phase of Gastro-CADx, features extracted in the first stage are applied to the discrete wavelet transform (DWT) and the discrete cosine transform (DCT). DCT and DWT are used to extract temporal-frequency and spatial-frequency features. Additionally, a feature reduction procedure is performed in this stage. Finally, in the third stage of the Gastro-CADx, several combinations of features are fused in a concatenated manner to inspect the effect of feature combination on the output results of the CADx and select the best-fused feature set. Two datasets referred to as Dataset I and II are utilized to evaluate the performance of Gastro-CADx. Results indicated that Gastro-CADx has achieved an accuracy of 97.3% and 99.7% for Dataset I and II respectively. The results were compared with recent related works. The comparison showed that the proposed approach is capable of classifying GI diseases with higher accuracy compared to other work. Thus, it can be used to reduce medical complications, death-rates, in addition to the cost of treatment. It can also help gastroenterologists in producing more accurate diagnosis while lowering inspection time.

## INTRODUCTION

Gastrointestinal (GI) disease is considered one of the supreme common diseases that usually infect people, causing complicated health conditions (*Du et al., 2019*). Based on the degree of injury, GI can approximately split into the precancerous lesion, primary GI cancer and progressive GI cancer, and benign GI diseases (*Sharif et al., 2019*). Among benign GI diseases are ulcers, gastritis, and bleedings which will not depreciate into cancers in short term. In contrast, precancerous GI injury could depreciate into primary GI cancer

Corresponding author
Omneya Attallah,
o.attallah@aast.edu

or even progressive GI cancer, in case it was not accurately diagnosed and treated in time (*Du et al., 2019*). Annually almost 0.7 million patients are diagnosed with gastric cancer. Since 2017, 135,430 new GI diseases arose in America. A global survey indicated that since 2017, 765,000 deaths occurred due to stomach cancer, 525,000 deaths are due to colon cancer. The poorest situations can be detected in the developing countries (e.g., the Asian countries and the Middle East) (*Ali et al., 2019*; *Khan et al., 2020a*). Moreover, among people diseased with GI diseases, 20% of them are from China, 18% from Brazil, 12% from Russia, 20% of EU, and 21% of the US (*Sharif et al., 2019*). The early diagnosis of GI is essential to reduce medical complications, cost of treatment, and lower death rates.

The traditional clinical method used for GI diagnosis is the intestinal biopsy of the GI tract. These biopsy samples are analyzed by medical experts using microscopes to examine the possibility of any cancerous or abnormal cells' existence. The drawbacks of such a method are being invasive and the necessity of a high degree of proficiency (*Ali et al., 2019*). In contrast, endoscopic imaging is a lower invasive technique for visualizing the GI tract (*Kainuma et al., 2015*). The endoscopic process assists the doctor in the recognition and diagnosis of gastric anomalies in their initial stages. Timely detection and diagnosis of chronic medical conditions can be healed with appropriate treatments. Hence, the imaging procedure can be very beneficial for a considerable decrease in medical complications, the cost of treatment, and death-rates, especially, the deaths that happen due to several GI cancers, which could be treated if cancer was discovered in its pre-malignant phase (*Hamashima et al., 2015*). Although there are numerous advantages in endoscopy, it brings along with it particular trade-offs; for example, the huge number of video frames produced during the screening process of the GI tract. On average, the entire process can take from 45 min to 8 h depending on the aimed GI region and the expertise of the gastroenterologist (*Ali et al., 2019*). The number of generated frames can reach up to 60,000 images. Most of these frames are redundant and not valuable and only a few images might have some abnormal lesions (*Khan et al., 2020b*). All these redundant images can be removed by examining each frame of the endoscopic video. Therefore, the manual examination of diseases through such a huge number of images is very challenging as it needs an extensive amount of time to observe the complete number of frames. Besides, at times the anomalous frames can be simply unnoticed by the gastroenterologist which can cause misdiagnosis. Therefore, such medical experts request automated schemes, that can automatically determine possible malignancies by analyzing the entire endoscopic images (*Aoki et al., 2019*).

Computer-aided diagnosis (CADx) are systems utilized for automatic diagnosis of several diseases within various parts of the human body like the brain (*Attallah, Sharkas & Gadelkarim, 2019*, *2020*), breast (*Ragab, Sharkas & Attallah, 2019*), lung (*Attallah, Ragab & Sharkas, 2020*), etc. Along with these diseases, CADx has been commonly used to diagnose GI disease in the intense by analyzing endoscopic images (*Khan et al., 2020b*). Such CADx has several advantages from which the patients, gastroenterologists, and medical students can benefit. These include; the reduction in the examination time of the whole endoscopic frames. Besides, the decrease in the cost of treatment as the lesion will be detected in an early phase. Moreover, CADx will improve the accuracy of the

diagnosis of GI diseases compared to manual examination. Also the inspection time from endoscopic images is to be decreased. Furthermore, it may be used for training medical staff and students without the necessity of an expert (*Ali et al., 2019*).

In a CADx scheme, the diagnosis is carried out using each frame depending on the significant features taken out from the image. Thus, feature extraction is the key step in an accurate diagnosis of medical conditions (*Attallah, 2020*) like GI diseases (*Khan et al., 2020b*; *Ali et al., 2019*; *Khan et al., 2019*). Several features are calculated using handcrafted techniques in the literature like color-based, texture-based, and some others (*Khan et al., 2020b*; *Ali et al., 2019*). *Karargyris & Bourbakis (2011)* utilized geometric and texture features extracted from SUSAN edge detector and Gabor filter extraction methods to detect small bowel polyps and ulcers. On the other hand, *Li & Meng (2012a)* used the uniform local binary pattern (LBP) and discrete wavelet transform (DWT). They employed an SVM classifier to detect abnormal tissues. In the same way, the authors in *Li & Meng (2012b)* detected tumors in the intestine using DWT and LBP. Instead, *Yuan & Meng (2014)* fuzed the saliency map with the Bag of Features (BoF) technique to identify polyps in endoscopic images. Initially, the authors employed the BoF method to describe the local features by using a scale-invariant feature transform (SIFT) feature vectors using k-means clustering. Next, the saliency map histogram method was utilized to extract salience features. Lastly, both features are combined and utilized to learn an SVM classifier. Later the same authors (*Yuan, Li & Meng, 2015*) added the complete LBP (CLBP), LBP, uniform LBP (ULBP), and histogram of oriented gradients (HoG) features along with SIFT features to extract additional distinctive texture features. Alternatively, color-based features were extracted in (*Ghosh, Fattah & Wahid, 2018*; *Deeba et al., 2018*) for bleeding detection.

Recently, the advancement of deep learning (DL) methods has delivered new opportunities to improve the analysis of endoscopic images. CNNs are the most type of networks used in endoscopy (*Alaskar et al., 2019*). These networks can be used as classifiers or/and feature extractors. Feature extraction methods based on DL techniques have been extensively utilized in the literature (*Ghatwary, Zolgharni & Ye, 2019*; *Kim, Cho & Cho, 2019*; *Lee et al., 2019*). The authors of *Khan et al. (2020a)* proposed a CADx system to detect ulcers and bleeding GI diseases. Their system extracted deep features from two different layers of VGG-16 CNN. Afterward, these features were fused, and then significant features were selected using an evolutionary search method called PSO. These features were then used to train the SVM classifier. *Igarashi et al. (2020)* proposed a CADx framework to classify several GI diseases using AlexNet. First, AlexNet extracted spatial features and then classified them into 14 different diseases. The authors of *Alaskar et al. (2019)* proposed a DL-based CADx that utilized AlexNet and GoogleNet for ulcer detection from low contrast endoscopic videos (WEV). Features extracted from these networks were classified using the fully connected layer of each network separately. AlexNet was also used in (*Fan et al., 2018*) to detect both erosions and ulcers that are observed in the intestine. *He et al. (2018)* introduced a framework based on two cascaded CNNs. The first network is VGG-16 CNN which was used for edge detection, whereas the second is the Inception CNN which was used for classification. Similarly,

*Khan et al. (2020b)* used two CNNs, the first one is Recurrent CNN for segmentation, whereas, the second was ResNet and was used for classification. The authors in *Yuan & Meng (2017)* suggested the use of an image manifold with stacked sparse auto-encoder to recognize polyps in endoscopic images. Instead, the authors in *Pei et al. (2017)* proposed a CADx system to recognize and assess the small bowel using features extracted from long short-term memory (LSTM).

Other research articles suggested the fusion of handcrafted features and DL features. *Sharif et al. (2019)* proposed a CADx system for classifying GI infections. The authors extracted deep features from VGG-16 and VGG-19 CNNs and fused these features with some geometric features. These fused features were then used as input to a K-nearest neighbors (KNN) classifier. Another system was presented in *Ghatwary, Ye & Zolgharni (2019)* to detect esophageal cancer. The system fuzed Gabor features and Faster Region-Based CNN (Faster R-CNN). On the other hand, *Billah, Waheed & Rahman (2017)* fuzed the color wavelet features and CNN features for detecting polyps. The combined features were used later to fed an SVM classifier. The authors in *Nadeem et al. (2018)* combined features extracted from textural analysis methods such as Haralick and LBP along with VGG-16 CNN DL features. The authors used logistic regression for classification. The authors of *Majid et al. (2020)* introduced a framework that combined the DCT, DWT, color-based statistical features, and VGG16 DL features for the recognition of several GI diseases. The authors used a genetic algorithm (GA) to select features using the KNN fitness function. Finally, the selected features were used to train an ensemble classifier. A summary of recent related work along with their limitations is shown in Table 1.

The main aim of this work is to construct a CADx called Gastro-CADx that is capable of accurately diagnosing more GI diseases than the proposed by others. Though there are various approaches to GI detection and classification in the literature, there exist some weaknesses among these methods which are summarized in table. Gastro-CADx tries to overcome the limitations found in related studies discussed in Table 1 through three cascaded stages. First of all, the majority of the current methods studied the detection and classification of a few types of GI anomalies, disease, or anatomical landmark. But, our proposed Gastro-CADx is an automatic highly accurate system to classify several GI diseases and anatomical landmarks. Some of the related studies are based on small dataset or used only one dataset to test the efficiency of their classification model, while Gastro-CADx is validated using two large datasets of several GI diseases. The few articles that classified several GI diseases achieved low accuracy, not reliable, or used only one type of CNN, whereas, Gastro-CADx is an accurate and reliable system that used more four CNNs. This article in the first stage, Gastro-CADx studies several CNN based methods for feature extraction from spatial domain instead of using one or two networks to benefit from the advantages of several types of CNNs. The previous studies were either based only on an end-to-end deep learning which has very high computational cost, used only spatial features extracted from CNNs or only handcrafted feature extractions, but Gastro-CADx is not based only on spatial features, but temporal-frequency and spatial-frequency features using handcrafted feature extraction methods as well not only

**Table 1 A summary of recent related studies.**

| Article | Purpose | Class | Method | Accuracy (%) | Limitation |
|---|---|---|---|---|---|
| *Khan et al. (2020b)* | Ulcer, polyp, bleeding detection | 4 | RCNN, ResNet101, and SVM | 99.13 | • Used only spatial features.<br>• Low segmentation accuracy for the ulcer regions.<br>• Fail for the segmentation of polyp and bleeding regions. |
| *Khan et al. (2020a)* | Ulcer, and bleeding detection | 3 | VGG-16, PSO, and SVM | 98.4 | • Limited classes<br>• Used only spatial features.<br>• High computational cost |
| *Igarashi et al. (2020)* | Classify several GI diseases | 14 | AlexNet | 96.5 | • Used only spatial features.<br>• The training or test data included chosen images of gastric cancer lesions, which could cause a selection bias.<br>• Has high computational cost<br>• Cannot be used in real-time examinations |
| *Alaskar et al. (2019)* | Ulcer detection | 2 | AlexNet & Google Net | 97.143 | • Limited classes.<br>• Used only spatial features |
| *Owais et al. (2019)* | Classification of multiple GI diseases | 37 | ResNet-18 and LSTM | 89.95 | • High computational cost.<br>• Used individual type of features<br>• Low accuracy |
| *Fan et al. (2018)* | Ulcer and Erosion detection | 2 | AlexNet | 95.16 95.34 | • Limited classes.<br>• Used only spatial features.<br>• Used only one type of CNN features<br>• The CADx was applied separately for ulcer and Erosion detection |
| *He et al. (2018)* | Hookworm detection | 2 | VGG-16 and Inception | 88.5 | • Limited classes.<br>• Used only spatial features<br>• Low accuracy |
| *Yuan & Meng (2017)* | Polyps detection | 2 | Stacked sparse auto-encoder with image manifold | 98 | • Limited classes.<br>• Used only spatial features |
| *Pei et al. (2017)* | Bowel detection and assessment | 2 | LSTM and PCA | 88.8 | • Limited classes.<br>• Used only temporal features.<br>• Used only one type of CNN features<br>• Low accuracy<br>• Small dataset |

(Continued)

| Article | Purpose | Class | Method | Accuracy (%) | Limitation |
|---|---|---|---|---|---|
| *Sharif et al. (2019)* | Ulcer, and bleeding detection | 3 | VGG-16, VGG-19, geometric features, KNN | 99.42 | • Limited classes.<br>• Small dataset.<br>• Used spatial and geometric features only |
| *Ghatwary, Ye & Zolgharni (2019)* | Esophageal cancer detection | 2 | Gabor Filter. faster R-CNN, and SVM | 95 | • Limited classes.<br>• Used only one type of CNN features<br>• Used spatial and textural based -Gabor features only.<br>• High computational cost |
| *Billah, Waheed & Rahman (2017)* | Polyps detection | 2 | Color based DWT, CNN, and SVM | 98.65 | • Limited classes.<br>• Used only one type of CNN features<br>• Used spatial and color based –DWT only<br>• Small dataset |
| *Nadeem et al. (2018)* | Classification of several GI diseases | 8 | VGG-19, Haralick and LBP texture analysis, and logistic regression | 83 | • Low accuracy<br>• Used only one type of CNN features<br>• Used spatial features based on CNN and textural analysis only |
| *Majid et al. (2020)* | Bleeding, esophagitis, polyp, and ulcerative-colitis classification | 5 | DCT, color based statistical features, DWT, VGG-16, GA, and E | 96.5 | • High computational cost.<br>• Used only one type of CNN DL features |
| *Nguyen et al. (2020)* | Classifying images to normal and abnormal | 2 | DenseNet, Inception, and VGG-16 | 70.7 | • Classify images to either normal or abnormal.<br>• Did not classify several GI diseases.<br>• Low accuracy |
| *Owais et al. (2020)* | Classification of multiple GI diseases | 37 | DenseNet and LSTM | 95.75 | • High computational cost.<br>• Used individual type of features |

end-to-end based DL. This appears clearly in the second stage of Gastro-CADx. It extracts handcrafted features based on textural analysis from the temporal-frequency and spatial-temporal domains using the DL features extracted in the first stage. This reduces the high computational cost of end-to-end DL techniques. Previous related studies indicated that CNN representations have improved the performance and the abstract level for the automatic detection and classification of GI diseases (*Majid et al., 2020*; *Khan et al., 2020b*; *Yuan & Meng, 2017*). Nevertheless, the fusion of CNN features with handcrafted variables could enhance diagnostic accuracy (*Majid et al., 2020*;

*Shi et al., 2018*). Therefore, in the third stage, a fusion process is introduced which combines the second stage features to benefit from the spatial, temporal- frequency, and spatial-frequency features. This stage can confirm the capacity of every feature abstraction method to mine significant information that might be disregarded from the other method. It can also reduce the computational cost compared to end-to-end DL methods.

The previous contributions are summarized to:

- Proposing an automatic and accurate CADx system called Gastro-CADx based on three stages to classify several GI diseases and anatomical landmarks.
- The system is not based only on spatial features, but temporal-frequency and spatial-frequency features using handcrafted feature extraction methods as well.
- In the first stage, Gastro-CADx studies several CNN based methods for feature extraction from spatial domain instead of using one or two networks to benefit from the advantages of several types of CNNs.
- In the second stage, Gatro-CADx extracts handcrafted features based on textural analysis from the temporal-frequency and spatial-temporal domains using the DL features extracted in the first stage.
- Also, in the second stage, Gastro-CADx tries to minimize the problem of computational time using only reduced dimensions of features.
- In the third stage, a fusion process is introduced which combines the second stage features to benefit from the spatial, temporal-frequency, and spatial-frequency features.
- The third stage can confirm the capacity of every feature abstraction method to mine significant information that might be disregarded from the other method.
- Gastro-CADx is validated using two large datasets of several GI diseases.
- Creating an accurate automatic diagnostic system that is reliable compared to related CADx systems.

## MATERIALS AND METHODS

### Dataset description

This article employs two datasets to evaluate the performance of Gastro-CADx.
The first dataset used in this article is called Kvasir (*Pogorelov et al., 2017*), and denoted as Dataset I. It consists of 4,000 images containing eight different GI classes. Three classes demonstrating anatomical landmarks, three demonstrating pathological states, and two associated with lesion-removal. The three anatomical landmark categories are pylorus, z-line, and cecum. The three diseased states are esophagitis, polyps, and ulcerative colitis. The two classes associated with lesion removal are dyed lifted polyps and dyed resection margins. The images are of different sizes from $720 \times 576$ up to $1,920 \times 1,072$ pixels. Some of these images include a green region illustrating the location and shape of the endoscope within the intestine. This information may be significant for later investigations (thus included) but must be wielded with care for the detection of the endoscopic findings. Figure 1 shows different image samples of different GI diseases.

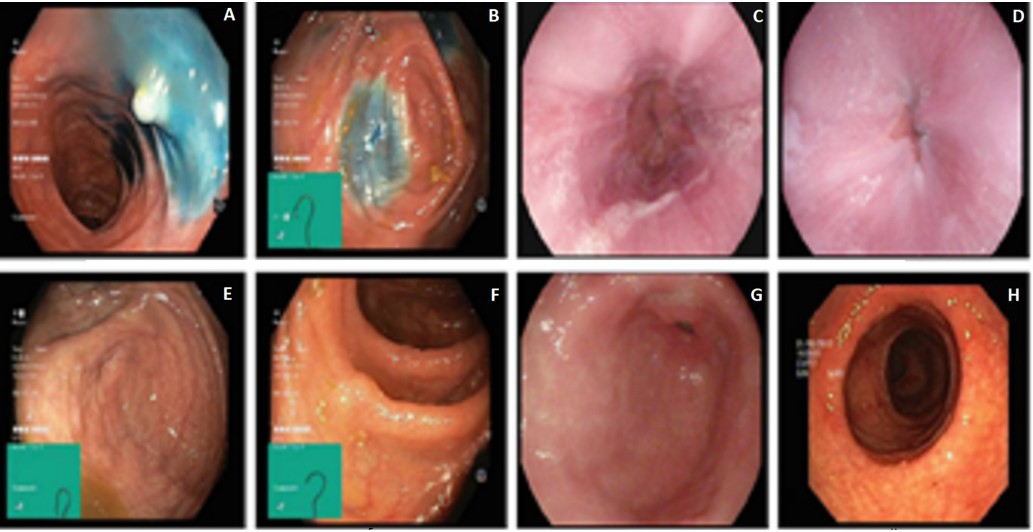

**Figure 1 Image samples of KASVIR dataset: (A) Dyed-Lifted-Polyp, (B) Dyed-Resection-Margin, (C) Esophagitis, (D) Normal-z-line, (E and F) Normal-Cecum, Polyps, (G) Normal-Pylorus, and (H) Ulcerative-Colitis.** Image credit: Michael Riegler.

The second dataset is called HyperKvasir (*Borgli et al., 2020*) and named Dataset II. The images and videos of this dataset were acquired using standard endoscopy equipment from Olympus (Olympus Europe, Hamburg, Germany) and Pentax (Pentax Medical Europe, Hamburg, Germany) at a Norwegian hospital from 2008 to 2016. The dataset consists of 10,662 labeled images and 23 classes. These classes are unbalanced; therefore, we chose only 10 balanced classes to construct Gastro-CADx. Four classes demonstrating anatomical landmarks, three demonstrating pathological states, one demonstrating quality of mucosal views, and two associated with lesion-removal. The three anatomical landmark categories are pylorus, z-line, pylorus, and cecum. The three pathological states are esophagitis, polyps, and ulcerative colitis. The two classes associated with lesion removal are dyed lifted polyps and dyed resection margins. The one demonstrating the quality of mucosal views is bowel quality. Figure 2 shows samples of images included in the dataset.

## Deep convolutional neural networks architectures

The popular type of DL approaches that is generally utilized for solving image-related classification problems in the health informatics field is the convolutional neural network (CNN) (*Jin et al., 2020*). In this article, four CNNs are utilized including; AlexNet, ResNet-50, DarkNet-19, and DenseNet-201 constructions. As it can be noticed from Table 1 that most related studies used AlexNet, ResNet and VGG CNNs. We did not use VGG as it has very high computational cost and number of parameters. Also, the features extracted from this network is of very huge size (*Bi et al., 2020*; *Ertosun & Rubin, 2015*; *Su et al., 2020*). Although, AlexNet is one of the oldest architectures, but it is still being used due to its acceptable performance. This is because it has efficient computation ability and

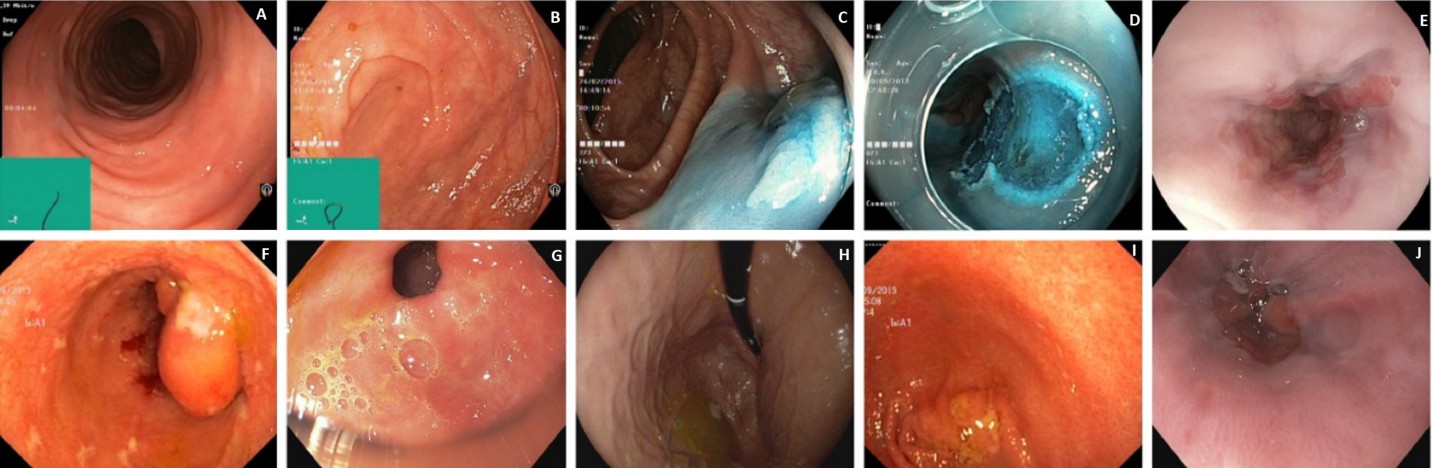

**Figure 2** **Image samples of Hyperkvasir dataset.** (A) Bowel quality, (B) Normal Cecum, (C) Dyed-Lifted-Polyp, (D) Dyed-Resection-Margin, (E) Esophagitis, (F) Polyps, (G) Polyrous, (H) Retroflex stomach, (I) Ulcerative-Colitis, and (J) Normal-z-line.

**Table 2** **A summary of the four CNNs *architectures*.**

| CNN Structure | Number of layers | Size of input | Size of output |
| --- | --- | --- | --- |
| AlexNet | 23 | 227 × 227 | 4,096 |
| ResNet-50 | 50 | 224 × 224 | 2,048 |
| DarkNet-19 | 19 | 256 × 256 | 8 and 10 |
| DenseNet-201 | 201 | 224 × 224 | 1,920 |

performs well with color images like these used in this article well (*Wang, Xu & Han, 2019*). We employed more recent CNNs like DarkNet and DenseNet architectures. To our own knowledge darknet was not used in the literature, whereas, only few articles used DenseNet for classifying GI diseases but has several drawbacks in their proposed methods. Therefore, we used these two new CNNs architectures to test their performance and ability to classify multiple GI diseases from endoscopic images.

The size of the input and output layers of the four networks employed used in the proposed method is shown in Table 2.

### AlexNet

The structure of AlexNet CNN was presented in 2012 by *Krizhevsky, Sutskever & Hinton (2012)*. This construction won the ImageNet Large-Scale Visual Recognition Challenge in 2012. The structure of AlexNet includes 23 layers corresponding to 5 convolutional layers, 5 rectified linear unit (ReLu) layers, 2 normalization layers, 3 pooling layers, 3 fc layers, a probabilistic layer using softmax units, and a classification layer ending in 1,000 neurons for 1,000 categories (*Attallah, Sharkas & Gadelkarim, 2020*).

### DarkNet-19

DarkNet was first introduced in 2017 by *Redmon & Farhadi (2017)*. DarkNet-19 is a CNN that is utilized as the spine of YOLO-V2. It commonly employs 3 × 3 filters and pairs

the number of channels after each pooling stage. DarkNet-19 utilizes global average pooling to perform classifications in addition to $1 \times 1$ filters to reduce the feature demonstration between $3 \times 3$ convolutions. Batch Normalization is applied to regularize the classification model batch, make the training process more stable, and accelerate convergence. Darknet-19 consists of 19 convolutional layers and 5 max-pooling layers.

### ResNet-50

ResNet architecture was first introduced in 2016. The essential constructing block of the ResNet is the residual block which was suggested by *He et al. (2016)*. The residual block offers shortcuts associations within the convolution layers, which can assist the network to step some convolution layers at a time. In other words, the residual block recommends two choices, it may attain a set of functions on the input, or it can permit this stage. Therefore, ResNet construction is supposed to be more effective than other CNNs such as AlexNet and GoogleNet as stated in *Attallah, Sharkas & Gadelkarim (2020)*. In this study, ResNet-50 is used which consists of 49 convolutional layers and one fc layer.

### DenseNet-201

The latest research has revealed that CNNs can be significantly deeper, more accurate, and effective to learn when they consist of smaller connections between layers near the input and those adjacent to the output. This finding motivated *Huang et al. (2017)* to propose the Dense Convolutional Network (DenseNet). DenseNet joins every layer to each other layer in a feed-forward manner. While conventional CNNs with M layers have M connections—one amid every layer and its succeeding layer, DenseNet has $M(M + 1)/2$ straight connections. For every layer, the feature-maps of all previous layers are utilized as inputs, and its feature-maps are utilized as inputs into all following layers. DenseNet has numerous benefits such as their ability to lessen the vanishing-gradient issue, reinforce feature dissemination, boost feature reprocesses, and considerably decrease the number of parameters. In this article, DenseNet-201 is used which has 201 layers deep.

## Proposed Gastro-CADx

An efficient hybrid CADx system called Gastro-CADx is proposed to classify several GI classes from endoscopic images. Gastro-CADx involves three steps including, the image preprocessing step, followed by feature extraction, reduction and fusion step, and finally a classification step. Initially, several augmentation processes are utilized to raise the number of images in the datasets. Also, images are resized. In the feature extraction, reduction, and fusion step, three stages are performed to construct Gastro-CADx. In the first stage, valuable deep features are extracted from four CNNs including (ResNet-50, AlexNet, DenseNet-201, and DarkNet-19). In the second stage, two handcrafted features are used to extract features from the spatial DL features extracted in the first stage. These handcrafted features are textural analysis based features representing temporal-frequency and spatial-frequency features. The dimension of these extracted features is reduced in this stage. Afterward, is the third stage of the Gastro-CADx, where several reduced features are fuzed in a concatenated manner. Finally, is the classification step in which machine

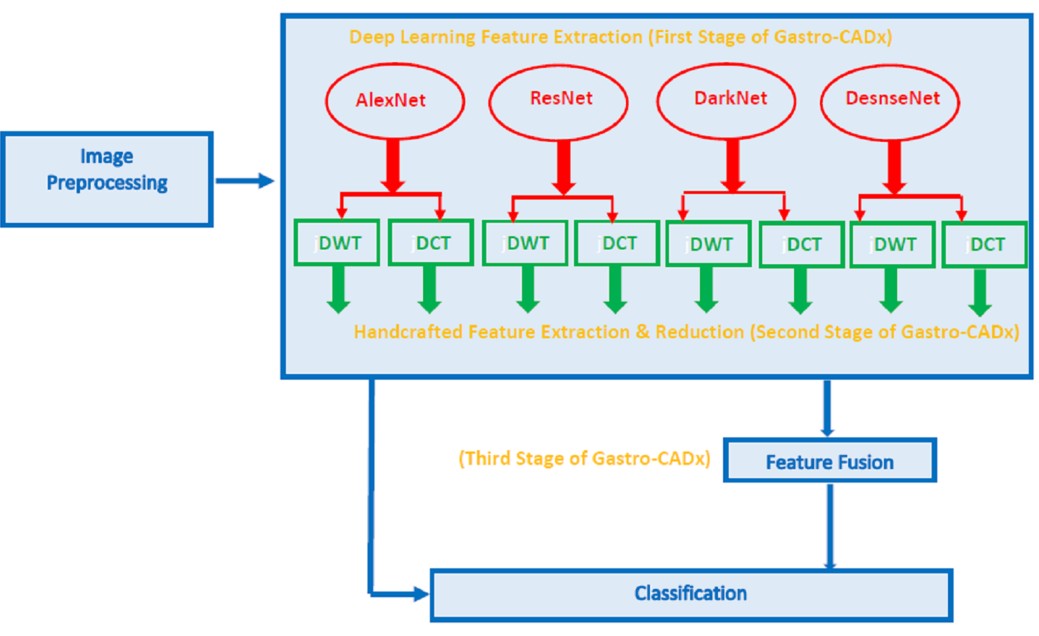

**Figure 3 The block diagram of the proposed Gastro-CADx.**

learning classifiers are used to identify several GI classes. Figure 3 represents the block diagram of Gastro-CADx.

### Image preprocessing step

The endoscopic images of both datasets are resized according to the size of the input layer of each CNN (*Attallah, Ragab & Sharkas, 2020*) shown in Table 2. Subsequently, these frames are augmented. The augmentation process is essential to raise the number of images (*Attallah, Sharkas & Gadelkarim, 2020*; *Ragab & Attallah, 2020*). This technique is performed because most likely the models which are learned with an insufficient quantity of frames may over-fit (*Ravì et al., 2016*; *Attallah, Sharkas & Gadelkarim, 2020*). The augmentation techniques utilized in this article to produce new endoscopic images from the training data are flipping, translation, transformation, and rotating (*Talo et al., 2019*; *Attallah, Ragab & Sharkas, 2020*). Each frame is flipped and translated in *x* and *y* directions with pixel range (−30, 30) (*Attallah, Ragab & Sharkas, 2020*). Furthermore, each endoscopic image is rotated with an angle range (0–180) degrees (*Ragab & Attallah, 2020*).

### Feature extraction, reduction, and fusion step

Gastro-CADx is based on three stages. The first stage is the DL feature extraction stage. The second is the handcrafted feature extraction and the reduction stage. Finally is the third stage known as the fusion stage.

### Deep learning feature extraction stage (First stage of Gastro-CADx)

Pre-trained CNNs trained using the endoscopic frames are used to accomplish feature extraction or classification processes. During the feature mining process, valuable DL features are mined from the CNNs. Instead of utilizing the CNNs for classification, DL

variables are pulled out from the fully connected layer called "fc7" as in *Attallah, Ragab & Sharkas (2020)*, the "global average pooling 2D layer" (fifth pooling layer), and the last average pooling layer of the AlexNet, ResNet-50, DarkNet, and DenseNet constructions as in (*Ragab & Attallah, 2020*). The DL features size are 4,096, 2,048, 8 or 10, and 1,920 for AlexNet, ResNet-50, DarkNet-19, and DenseNet-201 respectively.

*Handcrafted feature extraction and reduction stage (Second stage of gastro-CADx)Handcrafted feature extraction*

In this stage, time-frequency and spatial-frequency features based on textural analysis are determined from the DL features extracted in the previous stage. The textural features include the coefficients of the discrete wavelet transform (DWT) and the discrete cosine transform (DCT). Each feature method is discussed below. We employed DWT and DCT as they are popular feature extraction method based on textural analysis. One of the main benefit of DCT is its capability to spatially alter to characteristics of an image for instance discontinuities and changing frequency manner (*Bennet, Arul Ganaprakasam & Arputharaj, 2014*). It offers time-frequency representation of an image. Also, DCT has several advantages, first of all it prevents complicated calculation and presents simplicity of execution in practical purposes. Furthermore, DCT is capable of effectively managing the phase removing problem and demonstrates a powerful energy compaction estate (*Imtiaz & Fattah, 2010*; *Rashidi, Fallah & Towhidkhah, 2012*). DWT and DCT are the utmost common approach to extract textural features in the medical image processing area (*Lahmiri & Boukadoum, 2013*; *Srivastava & Purwar, 2017*; *Mishra et al., 2017*; *Anthimopoulos et al., 2014*; *Benhassine, Boukaache & Boudjehem, 2020*). Textural analysis based methods are useful in extracting texture features from images which is equivalent to simulating human visual learning procedure. It is widely used in medical image processing (*Attallah, 2021*; *Lahmiri & Boukadoum, 2013*; *Anwar et al., 2020*; *Castellano et al., 2004*).

- **Discrete wavelet transform (DWT)** is a widely used feature extraction method. DWT examines both signals and images (*Lahmiri & Boukadoum, 2013*; *Srivastava & Purwar, 2017*). It offers a temporal-frequency representation of an image or signal through decomposing them with the help of a group of orthogonal basis functions (Ortho-normal). Images are of two dimensions; therefore 2-D DWT is used to decompose the image (*Attallah, Sharkas & Gadelkarim, 2019*). One dimensional DWT is employed on each DL feature set distinctly which results in four groups of coefficients (*Ragab & Attallah, 2020*). The four groups generated after the 1-D DWT are known as three detail coefficients, $CD_1$, and approximation coefficients, $CA_1$. Detail coefficients consist of the diagonal, vertical, and horizontal coefficients, correspondingly.
- **Discrete Cosine Transform (DCT)** is frequently used to transform images into basic frequency components. It displays data as a sum of cosine functions oscillating at different frequencies (*Aydoğdu & Ekinci, 2020*). Generally, the DCT is applied to the imaged features to attain the DCT coefficients. The DCT coefficients are separated into three sets, known as low frequencies called (DC coefficients), middle frequencies, and high frequencies called (AC coefficients). High frequencies characterize noise and

small deviations (details). Whereas, low frequencies are associated with the brightness conditions. On the other hand, middle frequencies coefficients comprise valuable information and build the basic assembly of the image. The dimension of the DCT coefficient matrix is identical to the input DL featue (*Dabbaghchian, Ghaemmaghami & Aghagolzadeh, 2010*).Feature reduction

Feature reduction is an important procedure that is commonly used in the medical field to lower the huge dimension of the feature space. This reduction will correspondingly lead to a reduction in the complexity of the classification procedure (*Ragab & Attallah, 2020*), the training time of the model, and avoid overfitting (*Attallah et al., 2017a*; *2017b*). For this reason, DWT and DCT have been employed as feature reduction procedures as well as feature extractors instead of directly using the large dimension of DL features generated in the previous step. Therefore, a 1-level of DWT is applied to each DL features. The coefficients generated are are the approximation coefficients $CA_1$, and detail coefficients $CD_1$ of the first decomposition level of DWT. These coefficients have half the dimension of the original DL feature dimension which enters the DWT process. By this way the dimension of feature space is reduced. The CA and CD coefficients are used separately to train the SVM classifiers of the next step of Gastro-CADx.

The DCT, on its own, does not reduce the data dimension; however, it shrinks most of the image information in a small number of coefficients (*Dabbaghchian, Ghaemmaghami & Aghagolzadeh, 2010*). Another reduction stage is usually executed to reduce the data dimension, where some of the coefficients are chosen to develop feature vectors. In this article, 500 DCT coefficients are generated using the zigzag procedure. After this reduction procedure, these coefficients are used separately to train the SVM classifiers of the next step of Gastro-CADx.

*Feature fusion (Third stage of gastro-CADx)*

The feature vectors generated for each of the DCT and DWT coefficients are then fuzed in a concatenated manner to form different combinations of fuzed features sets which are then used to classify the SVM classifiers in the next step of Gastro-CADx. For DWT, initially, the CA coefficients extracted from the DL features for every two networks are fuzed. Then, the CA coefficients extracted from the DL features of every three networks are fuzed. Next, all CA coefficients extracted from DL features of the four networks are merged. The same procedure is done for the CD coefficients. For the DCT, firstly the coefficients extracted from the DL features for every two networks are fuzed. Then, the coefficients extracted from the DL features of every three networks are fuzed. Finally, the DCT coefficients extracted from DL features of the four CNNs are merged

**Classification step**

In this step, the classification procedure is performed using two scenarios either by an end-to-end DL (*Attallah, Ragab & Sharkas, 2020*), techniques or by using the features extracted from the three stages of Gastro-CADx. The scenarios resemble four experiments. The first scenario represents the use of the four CNNs including AlexNet, ResNet-50, DarkNet-19, and DenseNet-201 as classifiers (end to end DL process). Each pre-trained

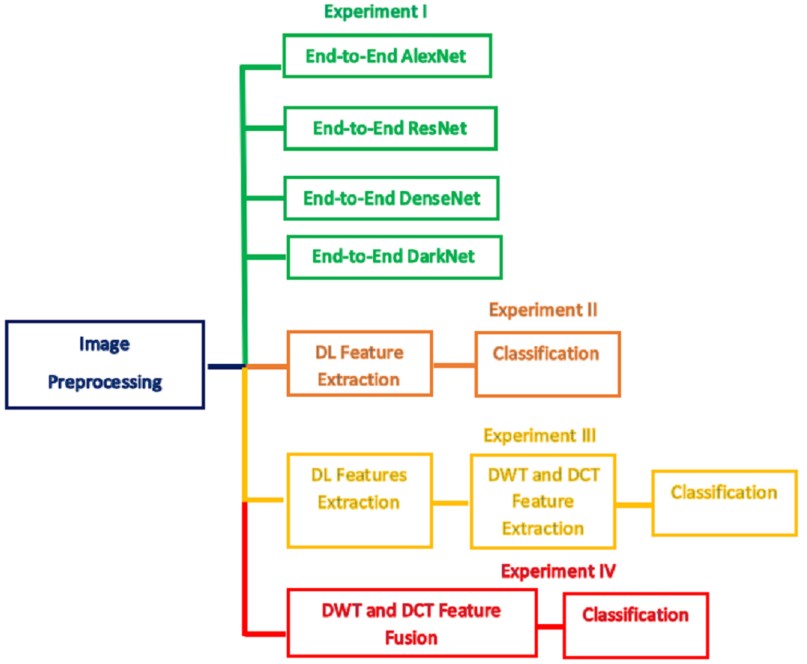

**Figure 4 A summary *of* the four experiments of Gastro-CADx.**

CNN is created and learned distinctly and then used as a classifier. The first scenario represents experiment I. In the second scenario, the first stage of Gastro-CADx is executed which corresponds to experiment II, where the pre-trained CNNs are applied to images, and then the DL features are extracted from each network individually. These DL features are used to learn distinct SVM classifiers. These features represent spatial information only and of a huge dimension. Therefore, in the second stage of Gastro-CADx which corresponds to experiment III, the DWT and the DCT feature extraction methods are applied to DL features generated from each CNN of the first stage of Gastro-CADx to extract temporal-frequency and spatial-frequency information. These features are utilized to train SVM classifiers individually. The problem of dimensionality reduction is considered as well in the second stage of Gastro-CADx, where a reduced set of coefficients are generated using the DWT and DCT methods. These coefficients represent feature vectors that are used separately to learn three SVM classifiers. Finally, in the third stage of Gastro-CADx, the reduced features are fuzed to form different combinations of fuzed features. These different combinations are used to construct several distinct SVM classifiers The aim of this stage is to examine the influence of feature fusion on the classification accuracy and select the combination which has the highest impact on the performance of the Gastro-CADx. This stage corresponds to experiment IV. Figure 4 summarizes the four experiments of Gastro-CADx.

Note that the SVM classifier was chosen as it is known to be a powerful classifier. It is considered to be one of the best methods in pattern classification and image classification (*Thai, Hai & Thuy, 2012*). It performs well with large dimension space and of multi-class

problems. As it uses kernel function which maps the feature space into new domain that can easily separate between classes of a dataset. Therefore, it is commonly used with the huge dimension of DL features extracted from CNNs (*Ragab et al., 2019*; *Jadoon et al., 2017*; *Zhang et al., 2020*; *Das et al., 2020*; *Xue et al., 2016*; *Leng et al., 2016*; *Wu et al., 2018*; *Sampaio et al., 2011*) achieving outperforming results. Also, as you can see in Table 1, that SVM is the commonly used in the literature It can be observed that articles that used SVM achieved the highest performance as *Khan et al. (2020b)* which achieved an accuracy of 99.13%, (*Khan et al., 2020a*) achieving an accuracy of 98.4%, (*Ghatwary, Ye & Zolgharni, 2019*) obtaining an accuracy of 95%, (*Billah, Waheed & Rahman, 2017*) achieving an accuracy of 98.65%

## EXPERIMENTAL SETUP

Several parameters are attuned after fine-tuning the fc layer of the CNNs. The number of epochs and the initial learning rate for the four CNNs are 10 and $10^{-4}$ respectively as in (*Attallah, Sharkas & Gadelkarim, 2020*). The mini-batch size and validation frequency are 10 and 3. The weight decay and momentum are set to $5 \times 10^{-4}$ and 0.9 respectively. The optimization algorithm used is the Stochastic Gradient Descent with Momentum (SGDM). To measure the capacity of the Gastro-CADx to classify several GI diseases, 5-fold cross-validation is engaged. This means that the GI datasets are divided into 80–20% for training and validation. The SVM classifiers are taught with 4 folds and verified by the remaining fold. Thus, the models are taught five times and the testing accuracy is calculated for each time then averaged. The kernel functions used for the SVM classifier are linear, quadratic, and cubic.

## EVALUATION PERFORMANCE

The presented Gastro-CADx framework is evaluated with numerous measures for instance; F1-score, precision, accuracy, sensitivity, and specificity. The formulas which are utilized in calculating such metrics are displayed below in Eqs. (1)–(5) (*Attallah, Ragab & Sharkas, 2020*).

$$\text{Accuracy} = \frac{\text{TP} + \text{TN}}{\text{TN} + \text{FP} + \text{FN} + \text{TP}} \tag{1}$$

$$\text{Sensitivity} = \frac{\text{TP}}{\text{TP} + \text{FN}} \tag{2}$$

$$\text{Specificity} = \frac{\text{TN}}{\text{TN} + \text{FP}} \tag{3}$$

$$\text{Precision} = \frac{\text{TP}}{\text{TP} + \text{FP}} \tag{4}$$

$$\text{F1} - \text{Score} = \frac{2 \times \text{TP}}{(2 \times \text{TP}) + \text{FP} + \text{FN}} \tag{5}$$

where; is the total sum of GI images that are well classified to the GI class which they actually belongs to is known as TP, TN is the sum of GI images that do not belong to the GI

**Table 3 The classification accuracy for the four CNNs used in Gastro-CADx using Dataset I.**

| CNN | Accuracy (%) |
| --- | --- |
| AlexNet | 88.32 |
| ResNet-50 | 91.66 |
| DarkNet-19 | 90.08 |
| DenseNet-201 | 89.83 |

**Table 4 The classification accuracy for the four CNNs used in Gastro-CADx using Dataset II.**

| CNN | Accuracy (%) |
| --- | --- |
| AlexNet | 91.66 |
| ResNet-50 | 94.75 |
| DarkNet-19 | 93.26 |
| DenseNet-201 | 91.93 |

class intended to be classified, and truly do not belong to. For each class of GI, FP is the sum of all images that are classified as this GI class but they do not truly belong to. For each class of GI, FN is the entire sum of GI images that are not classified as this GI class.

## RESULTS

The results of four experiments of Gastro-CADx are presented in this section. Experiment I is an end to end DL process where the four CNN are employed to perform classification. In experiment II (first stage of Gastro-CADx), DL features are extracted from the four CNNs and used to train distinct SVM classifiers. Experiment III (second stage of Gastro-CADx) represents the use of the second stage of feature extraction and reduction methods which employs DCT and DWT to extract temporal-frequency and spatial-frequency information from the images. In this experiment, reduced coefficients generated from DWT and DCT methods are employed to train SVM classifiers. In experiment IV, different combinations of fuzed features are generated and utilized to inspect the effect of feature combination on Gastro-CADx performance.

### Experiment I results

The results of the end-to-end DL procedure employed for classification are illustrated in Tables 3 and 4 for Dataset I and Dataset II respectively. Table 3 shows that the highest accuracy of 91.66% is achieved by ResNet-50 followed by an accuracy of 90.08%, 89.83%, 88.32% attained by DarkNet-19, DenseNet-201, and AlexNet respectively for Dataset I. Table 4 demonstrates that the peak accuracy of 94.75% is achieved by ResNet-50 followed by an accuracy of 93.26%, 91.93%,91.66% attained by DarkNet-19, DenseNet-201, and AlexNet respectively for Dataset II.

### Experiment II results

This experiment represents the first stage of Gastro-CADx. The results of this experiment are shown in Figs. 5 and 6 for Dataset I and Dataset II respectively. Figure 5 indicates the

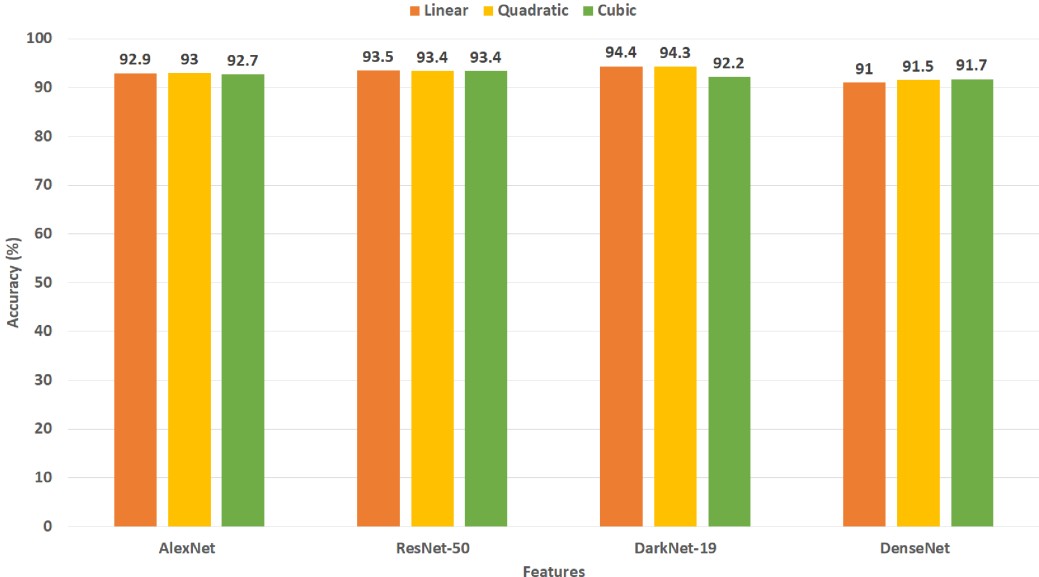

**Figure 5** Experiment II—Accuracy of each DL features extracted from the four CNNs of Gastro-CADx constructed using Dataset I.

maximum accuracies of 94.4% and 94.3% are attained by DarkNet-19 using linear and quadratic SVM classifiers using Dataset I. Subsequently, ResNet-50 features achieve an accuracy of 93.5%, 93.4%, and 93.4% using linear, quadratic, and cubic SVM classifiers respectively. Following, AlexNet and DenseNet-201 features obtain an accuracy of 92.9%, 93%, 92.7%, and 91%, 91.5%, 91.7% for using linear, quadratic, and cubic SVM classifiers respectively. Figure 6 shows the peak accuracies of 96.9%, 96.8%, 96.7% are achieved by ResNet-50 using linear, quadratic, and cubic SVM classifiers constructed with Dataset II. Next, DarkNet features attain an accuracy of 96.4%, 96%, and 95.2% using linear, quadratic, and cubic SVM classifiers respectively. Following, AlexNet and DenseNet-201 features obtain an accuracy of 95.5%, 95.7%, 95.3% and 94.7%, 94.6%, 94.6% for using linear, quadratic, and cubic SVM classifiers respectively.

## Experiment III results

This experiment represents the second stage of Gastro-CADx. The results of this experiment are shown in Figs. 7–10 for Dataset I and Figs. 11–14 for Dataset II. Figure 7 shows the classification accuracy for the three SVM classifiers constructed with CA and CD coefficients of DWT, besides the 500 DCT coefficients extracted from the ResNet-50 CNN using Dataset I. The figure indicates that the peak accuracy of 93.6% is achieved using the 500 DCT coefficients with linear SVM. Almost the same accuracy of 93.5% is attained using the CA coefficients of DWT.

Figure 8 demonstrates the classification accuracy for the three SVM classifiers built with CA and CD coefficients of DWT, in addition to the 500 DCT coefficients extracted from AlexNet CNN using Dataset I. The figure specifies that the highest accuracy of 93.3% is accomplished using the CD coefficients of DWT with a quadratic SVM classifier. A slightly lower accuracy of 92.9% is attained using the CA coefficients of DWT.

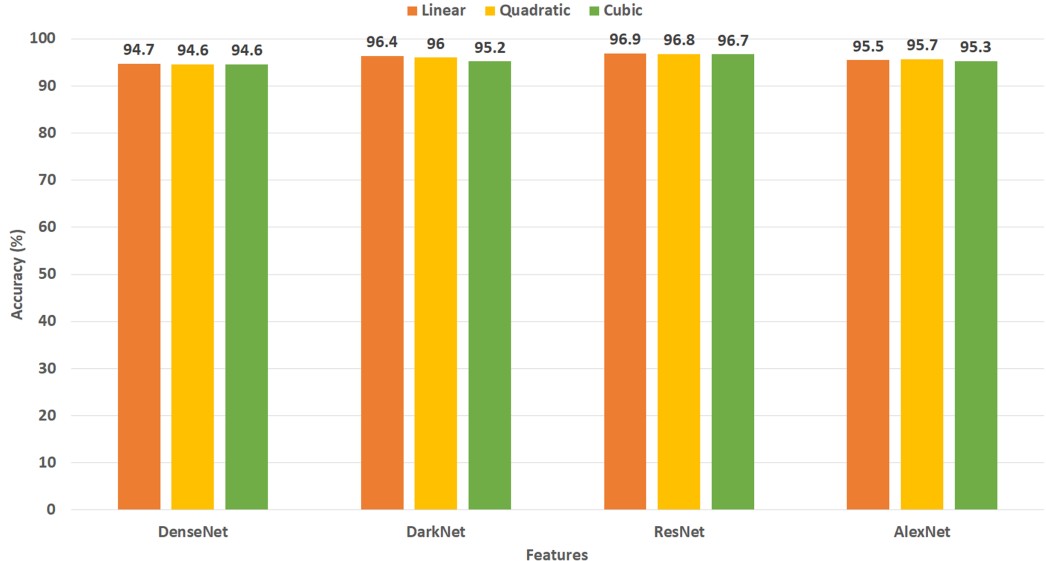

**Figure 6 Experiment II—Accuracy of each DL features extracted from the four CNNs of Gastro-CADx constructed using Dataset II.**

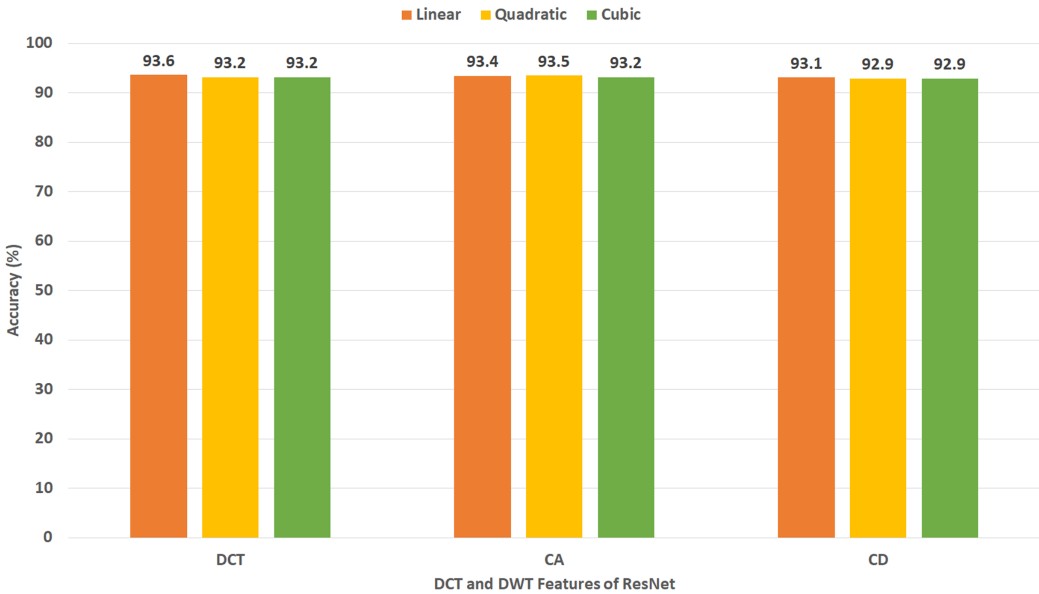

**Figure 7 Experiment III—Accuracy of each DCT and DWT features extracted from the ResNet-50 CNN of Gastro-CADx constructed using Dataset I.**

Figure 9 displays the classification accuracy for the three SVM classifiers constructed with CA and CD coefficients of DWT, as well as the 500, DCT coefficients extracted from DenseNet CNN using Dataset I. The figure identifies that the highest accuracy of 91.1% is accomplished using the CA coefficients of DWT with a cubic SVM classifier. A lower accuracy of 90.6% is reached using the CA coefficients of DWT with a linear SVM classifier.

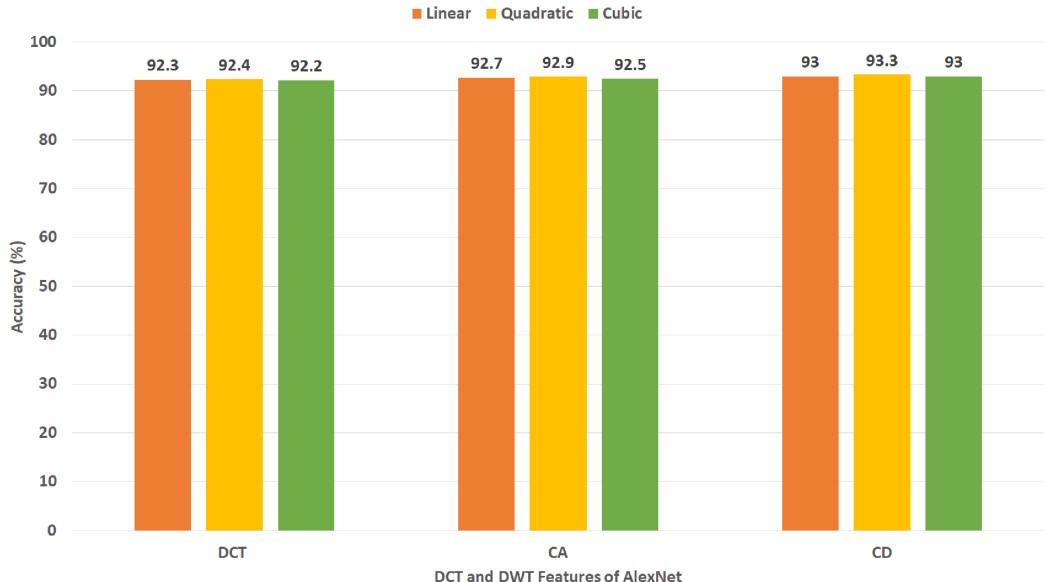

**Figure 8 Experiment III—Accuracy of each DCT and DWT features extracted from the AlexNet CNN of Gastro-CADx constructed using Dataset I.**

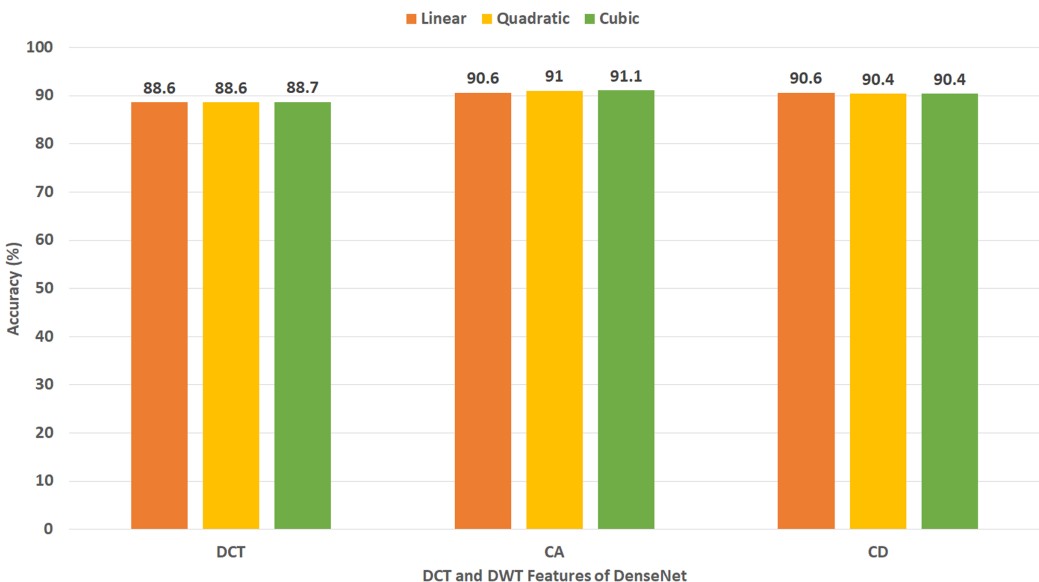

**Figure 9 Experiment III—Accuracy of each DCT and DWT features extracted from the DenseNet-201 CNN of Gastro-CADx constructed using Dataset I.**

Figure 10 shows the classification accuracy for the three SVM classifiers created with the CA and CD coefficients of DWT, besides the DCT coefficients extracted from DarkNet-19 CNN using Dataset I. Note that, since the number of DL features extracted from DarkNet-19 was only 8 (which is already a small dimension of features), all the DCT coefficients are used in this experiment without the need of the zigzag scanning procedure. The figure indicates that the highest accuracy of 94.7% is accomplished using the DCT coefficients with linear SVM.
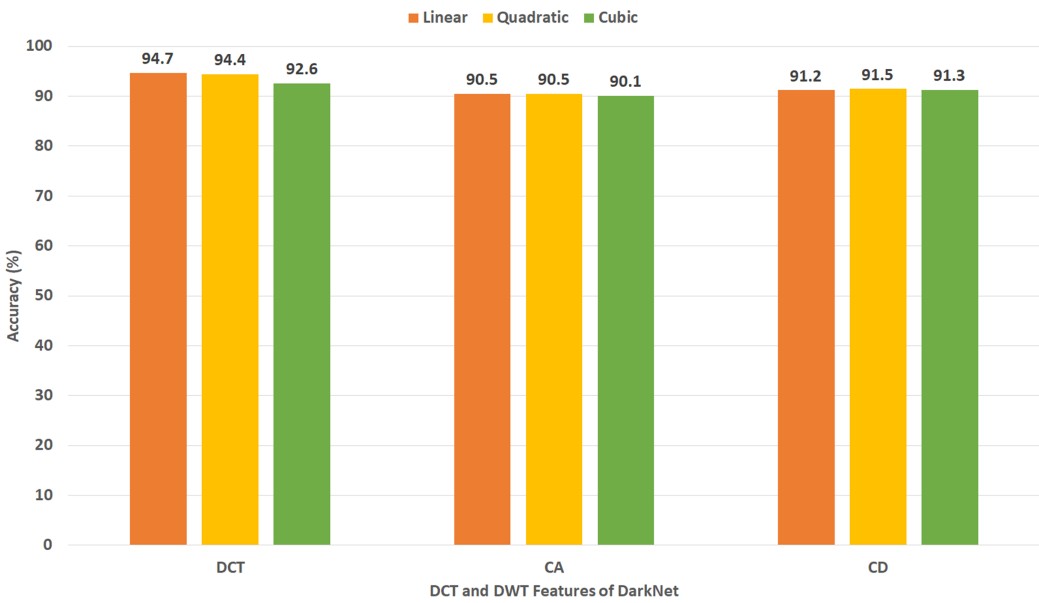

**Figure 10 Experiment III—Accuracy of each DCT and DWT features extracted from the DarkNet CNN of Gastro-CADx constructed using Dataset I.**

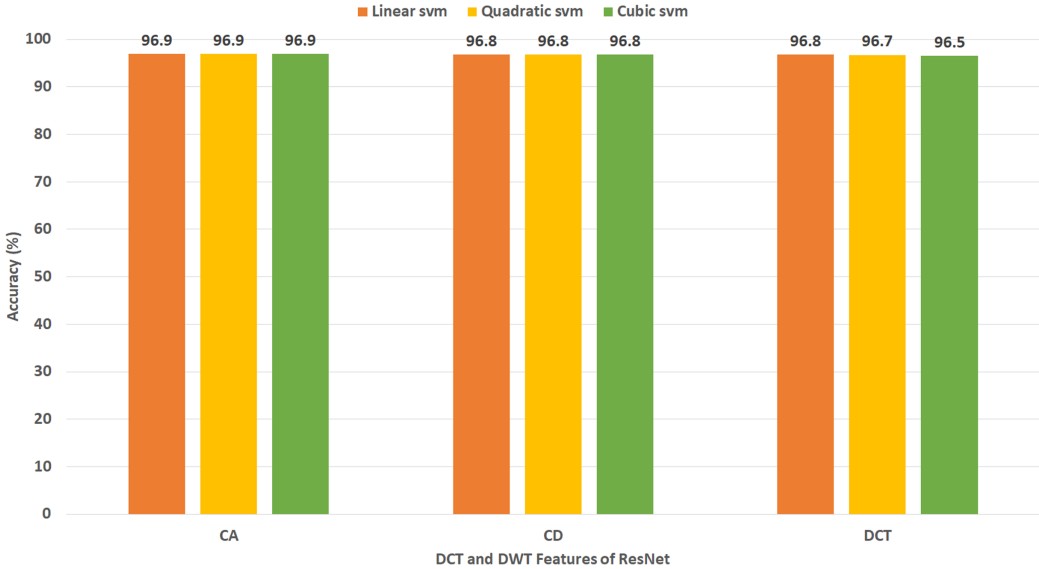

**Figure 11 Experiment III—Accuracy of each DCT and DWT features extracted from the ResNet-50 CNN of Gastro-CADx constructed using Dataset II.**

Figure 11 shows the classification accuracy for the three SVM classifiers constructed with CA and CD coefficients of DWT, besides the 500 DCT coefficients extracted from the ResNet-50 CNN using Dataset II. The figure indicates that the peak accuracy of 96.9% is achieved using the CA coefficients with linear, cubic, and quadratic SVM. Almost the same accuracy of 96.8% is attained using the CD coefficients of DWT with linear, cubic, and quadratic SVM and the 500 DCT coefficient with linear SVM.
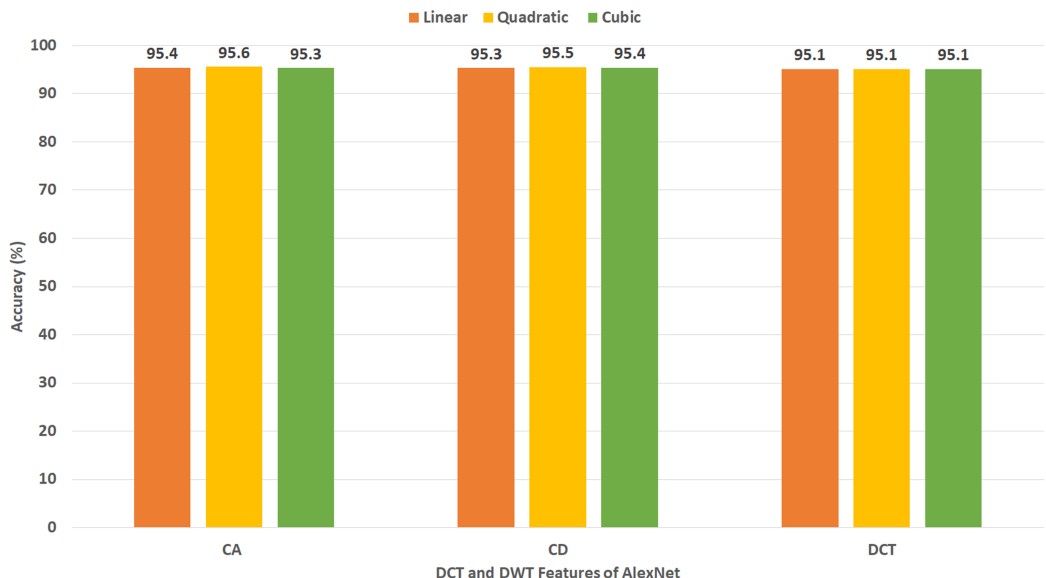

**Figure 12 Experiment III—Accuracy of each DCT and DWT features extracted from the AlexNet CNN of Gastro-CADx constructed using Dataset II.**

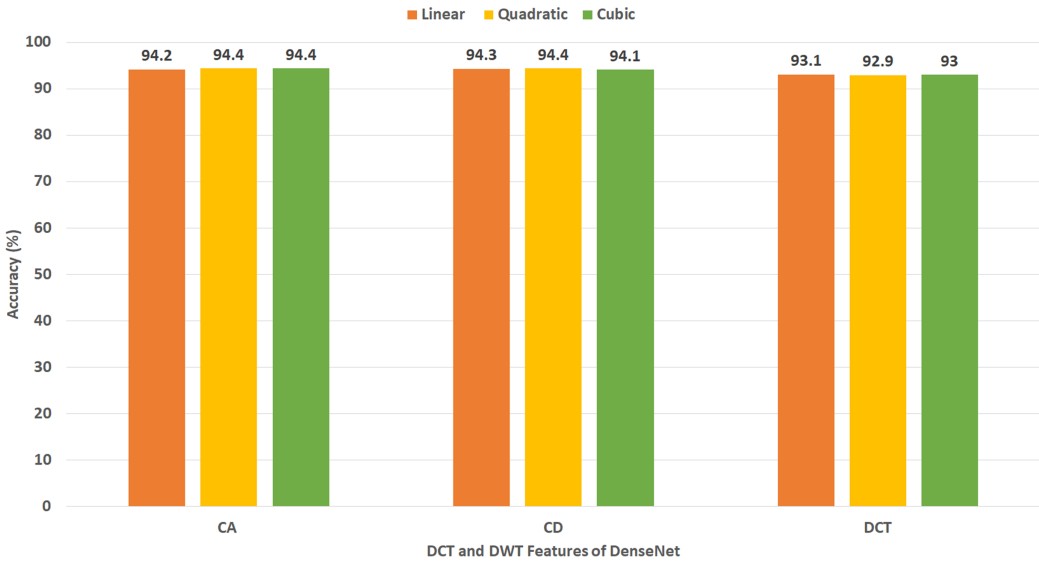

**Figure 13 Experiment III—Accuracy of each DCT and DWT features extracted from the DenseNet-201 CNN of Gastro-CADx constructed using Dataset II.**

Figure 12 reveals the classification accuracy for the three SVM classifiers learned with CA and CD coefficients of DWT, besides the 500 DCT coefficients extracted from AlexNet CNN using Dataset II. The figure specifies that the highest accuracy of 95.6% is accomplished using the CA coefficients of DWT with a quadratic SVM classifier. A slightly lower accuracy of 95.5% is attained using the CD coefficients of DWT with quadratic SVM.

Figure 13 indicates the classification accuracy for the three SVM classifiers built with CA and CD coefficients of DWT, besides the 500, DCT coefficients extracted from

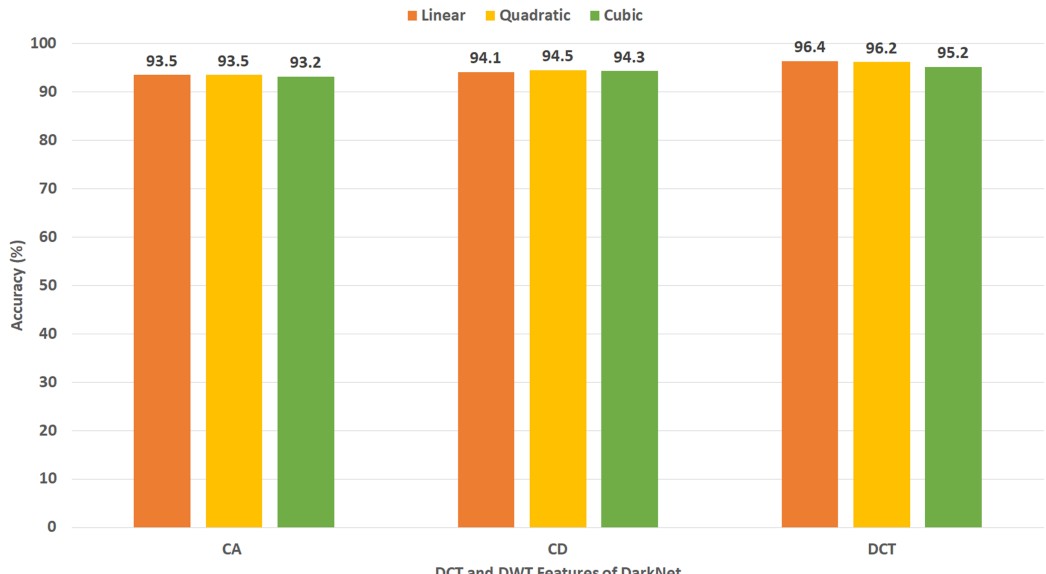

**Figure 14  Experiment III—Accuracy of each DCT and DWT features extracted from the DarkNet-19 CNN of Gastro-CADx constructed using Dataset II.**

DenseNet CNN using Dataset II. The figure identifies that the highest accuracy of 94.4% is accomplished using the CA coefficients of DWT with cubic and quadratic SVM classifiers. The same accuracy is reached using the CD coefficients of DWT with a quadratic SVM classifier.

Figure 14 demonstrates the classification accuracy for the three SVM classifiers constructed with CA and CD coefficients of DWT, in addition to the DCT coefficients extracted from DarkNet-19 CNN using Dataset II. As the number of DL features mined from DarkNet-19 was only 10 in the case of Dataset II (which is already a small dimension of features), all the DCT coefficients are employed in this experiment without the necessity of the zigzag scanning process. The figure specifies that the peak accuracy of 96.4% is obtained using the DCT coefficients with linear SVM.

## Experiment IV results

This experiment represents the third stage of Gastro-CADx. This experiment aims to explore the effect of combining features on the CADx's performance. Moreover, to search for the best combination of fuzed feature set which has the highest influence on the classification accuracy. To form the fuzed feature sets, firstly for DWT, the CD coefficients extracted from the DL features for every two CNNs are fuzed. Next, the CD coefficients extracted from the DL features of each three CNNs are merged. Afterward, all CD coefficients extracted from the DL features of the four CNNs are combined. A similar fusion process is executed for the CA coefficients. For DCT, initially, the coefficients extracted from the DL features for every two CNNs are fuzed. Afterward, the DCT coefficients extracted from the DL featured of each three CNNs are fuzed. Next, the coefficients extracted from DL images of the four CNNs are merged.

**Table 5 The classification accuracy (%) for the CA and CD features of DWT extracted from different combination of CNNs used in Gastro-CADx using Dataset I.**

| Features | CA | | | CD | | |
|---|---|---|---|---|---|---|
| | Linear | Quadratic | Cubic | Linear | Quadratic | Cubic |
| ResNet+DarkNet | 94.8 | 95 | 95 | 93.8 | 94 | 93.7 |
| AlexNet+DarkNet | 93.7 | 94.5 | 94.6 | 93.4 | 93.7 | 93.4 |
| DenseNet+DarkNet | 48.8 | 48.5 | 47.6 | 48.4 | 48.6 | 47.7 |
| AlexNet+ResNet | 94.2 | 94.4 | 94.1 | 94.3 | 94.7 | 94.5 |
| AlexNet+DenseNet | 95.7 | 96 | 96.2 | 95.8 | 96.2 | 96 |
| ResNet+DenseNet | 96.1 | 96.3 | 96.6 | 96.4 | 96.5 | 96.7 |
| AlexNet+ResNet+DenseNet | 96.8 | 97.3 | 97.3 | 96.5 | 96.8 | 96.8 |
| AlexNet+ResNet+DarkNet | 95.2 | 95.3 | 95.4 | 94.4 | 94.7 | 94.7 |
| AlexNet+DenseNet+DarkNet | 95.6 | 95.9 | 96.2 | 95.9 | 96.4 | 96.3 |
| ResNet+DenseNet+DarkNet | 96.3 | 96.7 | 96.7 | 96.3 | 96.4 | 96.5 |
| AlexNet+ResNet+DenseNet+DarkNet | 96.6 | 96.8 | 97 | 96.3 | 96.6 | 96.6 |

**Table 6 The classification accuracy (%) for the 500 DCT features extracted from different combination of CNNs used in Gastro-CADx using Dataset I.**

| Features | Linear | Quadratic | Cubic |
|---|---|---|---|
| ResNet+DarkNet | 95 | 95 | 95.3 |
| AlexNet+DarkNet | 95 | 95 | 95.1 |
| DenseNet+DarkNet | 47.7 | 47.5 | 46.6 |
| AlexNet+ResNet | 94.2 | 94.2 | 94.4 |
| AlexNet+DenseNet | 95.5 | 95.7 | 96.1 |
| ResNet+DenseNet | 96.2 | 96.2 | 96.3 |
| AlexNet+ResNet+DarkNet | 95.5 | 95.8 | 95.9 |
| AlexNet+ DenseNet +DarkNet | 95.8 | 95.9 | 96 |
| ResNet +DenseNet+DarkNet | 96.3 | 96.5 | 96.5 |
| AlexNet+ResNet+DenseNet | 96.8 | 97 | 97.1 |
| AlexNet+ResNet+DenseNet+DarkNet | 97.1 | 97.3 | 97.3 |

Table 5 displays a comparison between classification accuracy achieved using CA and CD features extracted from different combinations of the DL features generated from the four CNNs employed in Gastro-CADx using Dataset I. This comparison shows the CA features has slightly higher accuracy than CD features for all combination of fuzed features except for AlexNet+ResNet and AlexNet+DenseNet. The maximum performance (written in bold) is achieved using CA and CD features extracted using the fusion of AlexNet+ResNet+DenseNet CNNs, where the highest accuracy of 97.3% is attained using CA features extracted using AlexNet+ResNet+DenseNet CNNs with both quadratic and cubic SVM. On the other hand, Table 6 presents a comparison between classification accuracy accomplished using 500 DCT features extracted from different combinations of the DL variables produced from the four CNNs employed in Gastro-CADx using Dataset I.

**Table 7  The classification accuracy (%) for the CA and CD features of DWT extracted from different combination of CNNs used in Gastro-CADx using Dataset II.**

| Features | CA | | | CD | | |
|---|---|---|---|---|---|---|
| | Linear | Quadratic | Cubic | Linear | Quadratic | Cubic |
| ResNet+DarkNet | 98.6 | 98.6 | 98.6 | 98.4 | 98.6 | 98.6 |
| AlexNet+DarkNet | 98.6 | 98.7 | 98.9 | 98.1 | 98.6 | 98.7 |
| DenseNet+DarkNet | 97.1 | 97.3 | 97.3 | 96.5 | 96.7 | 96.7 |
| AlexNet+ResNet | 98.7 | 99 | 99 | 98.6 | 98.9 | 99 |
| AlexNet+DenseNet | 97.7 | 98 | 98.1 | 97.7 | 98 | 98.1 |
| ResNet+DenseNet | 48.8 | 49 | 49.2 | 48.7 | 49 | 49.1 |
| AlexNet+ResNet +DenseNet | 99.6 | 99.6 | 99.7 | 99.5 | 99.6 | 99.5 |
| AlexNet+ResNet+DarkNet | 98.8 | 98.8 | 98.9 | 98.8 | 98.9 | 98.9 |
| AlexNet+DenseNet+DarkNet | 98.7 | 98.7 | 98.8 | 98.6 | 98.8 | 98.8 |
| ResNet+DenseNet+DarkNet | 98.8 | 98.9 | 99 | 98.7 | 99 | 99.1 |
| AlexNet+ResNet+DenseNet+DarkNet | 99.6 | 99.6 | 99.6 | 99.4 | 99.5 | 99.6 |

This comparison indicates the maximum performance (written in bold) is achieved using DCT features extracted using AlexNet+ResNet+DenseNet+DarkNet CNNs, where the highest accuracy of 97.3% is attained using features extracted using AlexNet+ResNet+DenseNet+DarkNet CNNs with both quadratic and cubic SVM.

Table 7 demonstrates a comparison between the classification accuracy accomplished using CA and CD features extracted from different combinations of the DL variables produced from the four CNNs using Dataset II. This comparison indicates the CA features has slightly higher accuracy than CD features for all combinations of fuzed features except for AlexNet+ResNet and AlexNet+DenseNet. The peak performance (written in bold) is achieved using CA and CD features extracted using AlexNet+ResNet+DenseNet CNNs, where the maximum accuracy of 99.7% is reached using CA features extracted using AlexNet+ResNet+DenseNet CNNs using cubic SVM. In contrast, Table 8 shows a comparison between classification accuracy accomplished using 500 DCT features extracted from different combinations of the DL variables generated from the four CNNs employed in Gastro-CADx using Dataset II. This comparison specifies the maximum accuracy (written in bold) is achieved using 500 DCT features extracted using AlexNet+ResNet+DenseNet CNNs and AlexNet+ ResNet+DenseNet+DarkNet CNNs, where the highest accuracy of 97.3% is attained using linear, quadratic, and cubic SVM classifiers. Table 9 shows the performance metrics for cubic SVM classifiers trained with the fuzed CA features extracted from AlexNet+ResNet+DenseNet CNNs using Dataset I and Dataset II. The results of Table 9 indicate that the specificity of 0.9959 and 0.9996, sensitivity of 0.9715 and 0.9965, precision of 0.9718 and 0.9961, and F1 score of 0.9715 and 0.9963 are obtained for Dataset I and Dataset II respectively

**Table 8 The Classification Accuracy (%) for the 500 DCT features extracted from different combination of CNNs used in Gastro-CADx using Dataset II.**

| Features | Linear | Quadratic | Cubic |
|---|---|---|---|
| ResNet+DarkNet | 98.1 | 98.4 | 98.5 |
| AlexNet+DarkNet | 97.7 | 97.9 | 98 |
| DenseNet+DarkNet | 48.7 | 49 | 49.2 |
| AlexNet+ResNet | 98.2 | 98.5 | 98.6 |
| AlexNet+DenseNet | 98.2 | 98.6 | 98.6 |
| ResNet +DenseNet | 98.8 | 98.9 | 98.9 |
| AlexNet+ResNet+DarkNet | 98.8 | 99 | 99.2 |
| AlexNet+DenseNet+DarkNet | 98.6 | 98.7 | 98.7 |
| ResNet +DenseNet+DarkNet | 98.6 | 98.9 | 98.9 |
| AlexNet+ResNet +DenseNet | 99.6 | 99.6 | 99.6 |
| AlexNet+ResNet +DenseNet+DarkNet | 98.7 | 99.5 | 99.6 |

**Table 9 The Performance metrics for the CA features of DWT extracted from AlexNet+ResNet +DenseNet CNNs using Dataset I and II.**

| | Specificity | Sensitivity | Precision | F1 score |
|---|---|---|---|---|
| Dataset I | | | | |
| Cubic SVM | 0.9959 | 0.9715 | 0.9718 | 0.9715 |
| Dataset II | | | | |
| Cubic SVM | 0.9996 | 0.9965 | 0.9961 | 0.9963 |

## DISCUSSION

The manual diagnosis of GI diseases with a huge number of endoscopic images is very challenging and time-consuming. Besides, at times the image containing the abnormality can be simply unobserved by the medical expert which can lead to misdiagnosis. Therefore, there is an essential need for automatic systems that have the capability to automatically identify possible anomalies by analyzing the entire endoscopic images (*Aoki et al., 2019*). Nowadays, with the current development of DL and imaging processing technologies, CADx systems have been frequently used to help gastroenterologists in automatically examining endoscopic images and recognizing the GI disease (*Khan et al., 2020b*). In this study, an automatic CADx system called Gastro-CADx is proposed. The proposed CADx involves three steps including the image preprocessing step, followed by the feature extraction, reduction, and fusion step, and finally the classification step. Primary the endoscopic images were augmented. Next, is the feature extraction, reduction, and fusion step. Which presents the three stages of Gastro-CADx. In the first stage of Gastro-CADx, four spatial valuable DL features were extracted from the four CNNs and used to train SVM classifiers. Next, in the second stage of Gastro-CADx, DCT, and DWT feature extraction methods were employed to extract temporal-frequency and spatial-frequency features. These methods were used for feature reduction as well. These extracted features are utilized

**Table 10** The classification Accuracy (%) for the first stage of Gastro-CADx compared to Experiment I (end-to-end classifiction process) using Dataset I and II.

| CNN | Experiment I (END to END) | First Stage of Gastro-CADx | | |
| --- | --- | --- | --- | --- |
| | | Linear | Quadratic | Cubic |
| DataSet I | | | | |
| AlexNet | 88.32 | 92.9 | 93 | 92.7 |
| ResNet-50 | 91.66 | 93.5 | 93.4 | 93.4 |
| DarkNet-19 | 90.08 | 94.4 | 94.3 | 92.2 |
| DenseNet-201 | 89.83 | 91 | 91.5 | 91.7 |
| DataSet II | | | | |
| AlexNet | 91.66 | 95.5 | 95.7 | 95.3 |
| ResNet-50 | 94.75 | 96.9 | 96.8 | 96.7 |
| DarkNet-19 | 93.26 | 96.4 | 96 | 95.2 |
| DenseNet-201 | 91.93 | 94.7 | 94.6 | 94.6 |

to construct the SVM classifiers. Finally, in the third stage of Gastro-CADx, the coefficients of the DCT and DWT were fused to form different combinations of fuzed feature sets. This stage examined the influence of fuzing features on the performance of the CADx. Besides, the third stage of Gastro-CADx searched for the greatest mixture of features that influenced Gastro-CADx's performance. Two datasets, namely Dataset I and Dataset II were used to evaluate the performance of the proposed Gastro-CADx.

The first stage of Gastro-CADx is compared with the end-to-end DL CNNs of experiment I and the results are shown in Tables 10 for Dataset I and II. It can be observed from Table 10 that the first stage of Gastro-CADx has higher accuracies compared to the end-to-end CNNs constructed in experiment I for both datasets. The highest accuracy achieved in the first stage of Gastro-CADx is 94.4% using linear SVM trained with DarkNet-19 features for Dataset I (written in bold). Whereas, for Dataset II, the peak accuracy attained in the first stage of Gastro-CADx is 96.9% using linear SVM trained with DarkNet-19 features.

It was found that most of the previous studies directly used spatial DL features to perform the classification, however in the article we tried extracting spatial-temporal-frequency DL features using DWT and spatial-frequency DL features using DCT to examine their influence on the classification performance of Gastro-CADx (stage two of Gastro-CADx). DCT and DCT was also performed to reduce the huge dimension of the DL spatial features. It is proved from Fig. 15, that for dataset I, stage two has enhanced the classification performance with reduced feature set, while for dataset II it attained the same accuracy but with lower feature dimension. The second stage of Gastro-CADx has reduced the features extracted from the first stage of Gastro-CADx with almost the same accuracy but with fewer feature dimensional space for Dataset I and Dataset II. The highest accuracy of 94.7% of the second stage of Gastro-CADx for Dataset I was obtained using linear SVM classifier trained with the DCT coefficients extracted from deep learning features of DarkNet-19 CNN. Whereas, for Dataset II, the peak accuracy of 96.9%
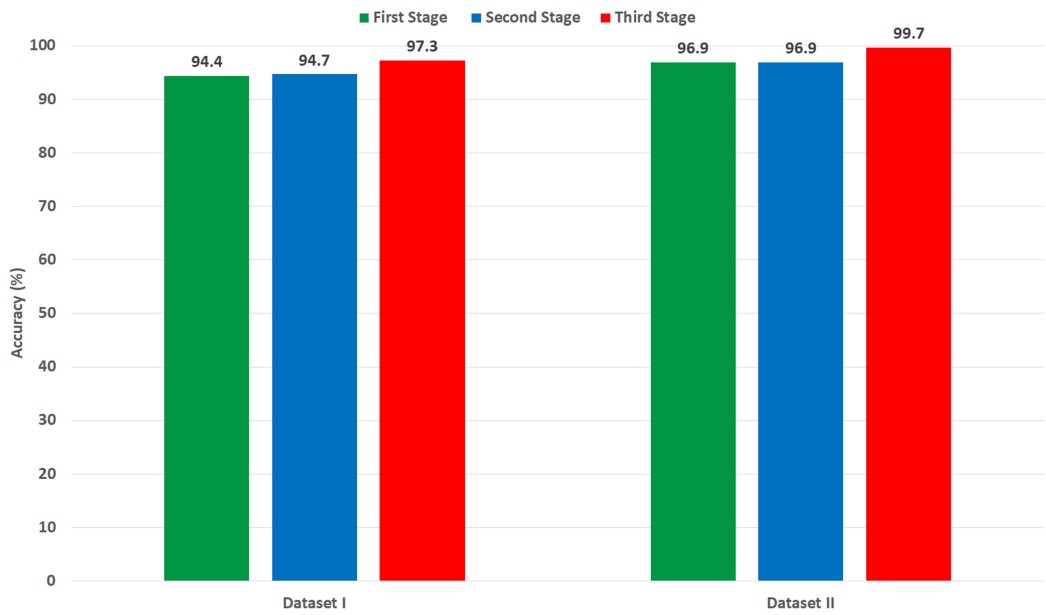

**Figure 15 A comparison between the highest accuracy attained from the three stage of Gasto-CADx using Dataset I and II.**                               

is achieved using linear SVM classifier trained with the CA coefficients extracted from deep learning features of ResNet-50 CNN.

On the other hand, the third stage of Gastro-CADx has further enhancement on the classification accuracy of Gastro-CADx as shown in Fig. 15 for Dataset I and Dataset II. Figure 15 shows the highest classification accuracy achieved using each stage of Gastro-CADx for Dataset I and II respectively. It can be noticed from the third stage of Gastro-CADx (experiment IV) that the fusion of DCT and DWT of DarkNet and DenseNet CNNs yielded the worst accuracy of around 47–49% for both Dataset I and Dataset II. Whereas, the highest accuracy of 97.3% and 99.7% is achieved using cubic SVM classifier trained with the fuzed CA coefficients extracted using deep learning features of AlexNet+ResNet+DesneNet for Dataset I and Dataset II respectively. In order to make a fair comparison regarding the computational time with other related studies, we both should use the same platform and environment like using the same processor and video controller and other specifications which can vary the computational time. Since, this is very hard to be accomplished as an alternative, we compared the computational cost of the proposed Gastro-CADx with ResNet CNN (end-to-end deep learning techniques) which is widely used in the literature and achieved the highest accuracy using both Dataset I and Dataset II as shown in Table 10. This comparison is shown in Table 11 which compares both the classification accuracy and training time of ResNet CNN using end-to-end procedure with Gastro-CADx. Table 11 proves that Gastro-CADx has much lower computation time than ResNet (end-end classification) while attaining higher accuracy for both datasets. This is because the computation time for ResNet is 80,580 s and 100,800 s for Dataset I and II respectively which is much higher than the 210 s and 780 s achieved by Gastro-CADx. Also, the accuracy for ResNet is 90.08% and 94.75% for

**Table 11 Computation time and accuracy acheived using Gastro-CADx compared to ResNet end-to-end deep learning method.**

| Method | Dataset I | | Dataset II | |
|---|---|---|---|---|
| | Training time (s) | Accuracy (%) | Training time (s) | Accuracy (%) |
| ResNet (end-to end) | 80,580 | 90.08 | 100,800 | 94.75 |
| Gastro-CADx | 210 | 97.3 | 780 | 99.7 |

**Table 12 Comparisons with recent related studies based on Dataset I.**

| Article | Method | Performance metrics | | | | |
|---|---|---|---|---|---|---|
| | | Accuracy (%) | Sensitivity | Precision | Specificity | F1 score |
| *Ahmad et al. (2017)* | AlexNet | 75.4 | – | – | – | – |
| *Agrawal et al. (2017)* | GF[1]+Inception V3+VGG+SVM | 96.1 | 0.852 | 0.847 | 0.978 | 0.827 |
| *Pogorelov et al. (2017)* | ResNet+LMT[2] | 95.7 | 0.826 | 0.829 | 0.975 | 0.802 |
| *Pogorelov et al. (2017)* | GF+Decision Tree | 93.7 | 0.748 | 0.748 | 0.964 | 0.747 |
| *Pogorelov et al. (2017)* | GF+Random Forest | 93.3 | 0.732 | 0.732 | 0.962 | 0.727 |
| *Nadeem et al. (2018)* | GF+LBP[3]+Haralick+LR[4] | 94.2 | 0.774 | 0.767 | 0.966 | 0.707 |
| *Thambawita et al. (2018)* | GF+CNN | 95.8 | 0.958 | 0.9587 | 0.9971 | 0.9580 |
| *Owais et al. (2019)* | ResNet+DenseNet+MLP[5] | 92.5 | 0.993 | 0.946 | – | 0.934 |
| Gastro-CADx | | 97.3 | 0.9715 | 0.9718 | 0.9959 | 0.9715 |

**Notes:**
[1] Global features.
[2] Logistic model tree.
[3] Local binary pattern.
[4] Logistic regression.
[5] Multilayer perceptron.

Dataset I and II respectively which is much higher than the 97.3% and 99.7% obtained by Gastro-CADx Note that we also searched related studies to see if the authors have mentioned the computational time of their proposed methods, but unfortunately this information was missing.

All experiments are done with Matlab 2020a. The processor used is Intel(R) Core(TM) i7-7700HQ, CPU @ 2.80 GHz 2.8 GHz, RAM 16 GB, and 64-bit operating system. The video controller is NVIDIA GeForce GTX 1050.

A comparison is made to compare the performance of Gastro-CAD with the latest relevant work that used Dataset I. The results of this assessment are displayed in Table 12. Table 12 results prove the competence of Gastro-CADx compared to other previous related studies. Gastro-CADx proposed in this article appears to perform well on all of the metrics provided in Table 12. Gastro-CADx outperformed the systems presented by *Ahmad et al. (2017)*, *Pogorelov et al. (2017)* (first method), *Owais et al. (2019)*, as they used only spatial information extracted and from one or two CNN. The proposed system also outperformed (*Pogorelov et al., 2017*; *Nadeem et al., 2018*) as they used only handcrafted global features and did not benefit from using the spatial information of features extracted with DL techniques. Although, *Agrawal et al. (2017)* combined DL features with handcrafted global features, yet their performance is lower than

**Table 13 Comparisons with studies based on Dataset II.**

| Method | Accuracy (%) |
| --- | --- |
| AlexNet | 91.66 |
| ResNet-50 (*Borgli et al., 2020*) | 94.75 |
| DarkNet-19 | 93.26 |
| DenseNet-201 | 91.93 |
| Gastro-CADx | 99.7 |

Gastro-CADx. This is because Gastro-CADx considered the fusion of two types of textural features while reducing the feature space.

Dataset II is a new dataset for GI disease that was just released in 2020. Therefore, there is still no research articles to compare with. For this reason, we only compared with the ResNet-50 CNN used in *Borgli et al. (2020)* as well as the other three CNNs employed in experiment I of Gastro-CADx and illustrated in Table 13. The results of Gasto-CADx shown in Table 12 verifies its competence. It outperformed the classification accuracy achieved by ResNet-50 used in *Borgli et al. (2020)*. Gastro-CADx has also better performance than the classification accuracy achieved by AlexNet, DenseNet-201, and DarkNet-19 CNNs. This is because Gastro-CADx extracted not only spatial features but temporal-frequency and spatial-frequency features. It also used DCT and DWT not only for feature extractors but also for feature reduction methods. Moreover, it fuzes these several reduced t features to enhance the performance of the CADx.

The three stages of Gatro-CADx based on deep CNNs, DCT, and DWT showed the best performance with the highest accuracies of 97.3% and 99.7% for Dataset I and Dataset II respectively. The following article (*Attallah, 2020*; *Colquhoun, 2014*), that that the reliability of a medical system requires that the sensitivity should be greater than or equivalent to 80%, the specificity is greater or equivalent to 95%, and the precision is more or equivalent to 95%. The specificities, sensitivities, and precision shown in Table 9 are all larger than 95%, therefore Gastro-CADx can be considered a reliable system. This remarkable reliability and performance of Gastro-CADx rises its usability in the diagnosis of several GI diseases by automatically detecting several types of GI lesions or anomalies. Our AI-based Gastro-CADx framework can help the medical experts in an effective diagnosis of several complex GI diseases. Furthermore, it may assist gastroenterologists in reaching a more accurate diagnosis whereas reducing examination time. The proposed system can be used to decrease medical obstacles, death-rates in addition to the cost of treatment.

## CONCLUSION

This article introduced a CADx system called Gastro-CADx for the automatic classification of GI diseases based on DL techniques. Gastro-CADx consist of three stages. The first stage is based on DL feature extraction techniques to extract spatial information from endoscopic images. The second stage extracted some temporal-frequency and spatial-frequency features. The feature reduction procedure is also considered in this stage.

The third stage is a feature fusion based process where several features sets extracted in the second stage are fuzed to form numerous combinations of fuzed features. The results of the three stages of Gastro-CADx verified that the proposed system was capable of accurately classifying GI diseases. The first stage of Gastro-CADx achieved higher accuracy than that of end to end DL CNNs. Moreover, the results of the second stage of Gastro-CADx indicated that using the temporal-frequency and spatial-frequency has a better performance compared to using only spatial features. Besides, the second stage of Gastro-CADx achieved competitive performance to the first stage with a lower dimension of features. Also, the third stage has further improvement in the performance of Gastro-CADx which indicated that feature fusion had a significant impact on the accuracy of classification. The performance of the Gastro-CADx is competitive with recent related work based on the same dataset. This means the proposed method can be used efficiently for the diagnosis and classification of GI diseases. Consequently, the cost of medical investigations, medical complications, and death-rates will be reduced. Moreover, the quality of diagnosis will be enhanced as well as the accuracy. Future work will focus on combining multiple datasets to form a multicenter study. Besides, exploring more CNNs and more handcrafted feature extraction methods.

### Funding
The authors received no funding for this work.

### Competing Interests
The authors declare that they have no competing interests.

### Author Contributions
- Omneya Attallah conceived and designed the experiments, performed the experiments, analyzed the data, prepared figures and/or tables, authored or reviewed drafts of the paper, and approved the final draft.
- Maha Sharkas performed the experiments, analyzed the data, performed the computation work, authored or reviewed drafts of the paper, and approved the final draft.

### Data Availability
The Kvasir dataset (kvasir-datasetV1) is available at *Pogorelov et al. (2017)* and Simula: https://datasets.simula.no/kvasir/.

The HyperKavasir dataset (labeled images) is also available at Borgli et al. (2019) (DOI 10.31219/osf.io/mkzcq) and Simula: https://datasets.simula.no/hyper-kvasir/.

### Supplemental Information
Supplemental information for this article can be found online at http://dx.doi.org/10.7717/peerj-cs.423#supplemental-information.

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
