# Peer review of "GASTRO-CADx: a three stages framework for diagnosing gastrointestinal diseases"

_PeerJ Computer Science, doi:10.7717/peerj-cs.423_

## Round 0.1 · original submission · Major Revisions

Follow the reviewer comments when revising the manuscript.

Reviewer 1 ·

Basic reporting

In general, this paper is well written, although part of it should be improved to ensure that readers can clearly understand the text. Some examples where the language could be improved are listed as following.
Line 74, How to get the results after "Therefore"? The statements before "Therefore" are not reasons for "the early diagnosis of GI is essential …".
Line 402, What does "… as as …" mean?
Lines 423-425, FP and FN are all the images' sum that is not well identified for each class of GI diseases. What is the difference between FP and FN?
Line 655, "stage two" should be written as "the second stage".

Literature references and sufficient field background have been provided. Moreover, this article is well structured. Figures are relevant to the content of the article. However, the resolution of Figures 5-14 is not sufficient. The raw data are available.

Experimental design

The research question is well defined, relevant, and meaningful. Moreover, the rigorous investigation has been performed to a high technical & ethical standard. Methods have been described with sufficient information.

However, the simulation environment is not mentioned. Furthermore, there are some inconsistent statements for the methods, and some information is not clear. Some examples are listed below.

Line 287, in the first stage, four CNN are used as feature extractors.
Lines 312-320, in the first stage, valuable DL features are mined from the CNNs.
Lines 573-575, in the first stage, four DL features were extracted from images and used to train SVM classifiers.
The problem is which statement describes the real first stage.

Line 336, What are CD1 and CA1?

Validity of the findings

Although the results and compelling, the data analysis does not include the experimental settings' meaning corresponding to the best results.

The conclusions are well stated and linked to the research question.

Reviewer 2 ·

Basic reporting

Comments are attached

Experimental design

Comments are attached

Validity of the findings

Comments are attached

Annotated reviews are not available for download in order to protect the identity of reviewers who chose to remain anonymous.

---

## Round 0.2 · accepted · Accept

The article is accepted. Congratulations!